# Potential Therapeutic Approaches Against Brain Diseases Associated with Cytomegalovirus Infections

**DOI:** 10.3390/ijms21041376

**Published:** 2020-02-18

**Authors:** Shao-Cheng Wang, Shiu-Jau Chen, Yuan-Chuan Chen

**Affiliations:** 1Jianan Psychiatric Center, Ministry of Health and Welfare, Tainan 71742, Taiwan; WShaocheng@gmail.com; 2Department of Mental Health, Johns Hopkins Bloomberg School of Public Health, Baltimore, MD 21205, USA; 3Department of Neurosurgery, Mackay Memorial Hospital, Taipei 10449, Taiwan; chenshiujau@gmail.com; 4Department of Medicine, Mackay Medicine College, New Taipei City 25245, Taiwan; 5Program in Comparative Biochemistry, University of California, Berkeley, CA 94720, USA

**Keywords:** cytomegalovirus, brain disease, mental retardation, glioblastoma, gene/RNA targeting approach, vaccine therapy, immunotherapy

## Abstract

Cytomegalovirus (CMV) is one of the major human health threats worldwide, especially for immunologically comprised patients. CMV may cause opportunistic infections, congenital infections, and brain diseases (e.g., mental retardation and glioblastoma). The etiology of brain diseases associated with human CMV (HCMV) infections is usually complex and it is particularly difficult to treat because HCMV has a life-long infection in its hosts, high mutation rate, and latent infections. Moreover, it is almost impossible to eradicate latent viruses in humans. Although there has been progress in drug discovery recently, current drugs used for treating active CMV infections are still limited in efficacy due to side effects, toxicity, and viral resistance. Fortunately, letermovir which targets the HCMV terminase complex rather than DNA polymerase with fewer adverse reactions has been approved to treat CMV infections in humans. The researchers are focusing on developing approaches against both productive and latent infections of CMV. The gene or RNA targeting approaches including the external guide sequences (EGSs)-RNase, the clustered regularly interspaced short palindromic repeats (CRISPR)/CRISPR-associated protein 9 (Cas9) system and transcription activator-like effector nucleases (TALENs) are being investigated to remove acute and/or latent CMV infections. For the treatment of glioblastoma, vaccine therapy through targeting specific CMV antigens has improved patients’ survival outcomes significantly and immunotherapy has also emerged as an alternative modality. The advanced research for developing anti-CMV agents and approaches is promising to obtain significant outcomes and expecting to have a great impact on the therapy of brain diseases associated with CMV infections.

## 1. Introduction

### 1.1. Brain Disease

The brain is one’s body control center which controls thoughts, memory, learning, speech, and movement, and regulates many organ functions. When the brain is healthy, it works quickly and automatically. However, when the brain is damaged, it can affect many different things such as sensation, memory, activity, and even personality. Brain diseases include any conditions or disabilities that affect the brain, and major types are brain injury, mental disorder, neurodegenerative disease, mental retardation, and brain tumor (Table 1, [1,2,3,4,5,6,7,8,9,10,11,12,13,14,15]).

### 1.2. Cytomegalovirus Overview

There are nine distinct human herpesvirus (HHV) species known to cause human diseases, including herpes simplex virus-1(HSV-1, HHV-1), herpes simplex virus-2 (HSV-2, HHV-2), varicella zoster virus (VZV, HHV-3), Epstein-Barr virus (EBV, HHV-4), cytomegalovirus (CMV, HHV-5), roseolovirus (HHV-6A, HHV-6B, HHV-7), and Kaposi’s sarcoma-associated herpesvirus (KSHV, HHV-8) [16,17]. CMV belongs to the genus of *Herpesvirus* in the family Herpesviridae (subfamily Betaherpesvirinae), and in the order Herpesvirales. Human cytomegalovirus (HCMV), the most studied CMV with a double-stranded DNA genome of about 230 kb, consists of envelope, tegument, capsid, and DNA core from outside to inside. Like all herpesviruses, HCMV encodes its genes in a highly coordinated transcriptional cascade and HCMV genes are expressed in an order as follows: (1) Immediate early genes, which encode regulatory proteins; (2) early genes, which encode enzymes for viral DNA replication; (3) late genes, which encode structural proteins for infectious viral particles (virions) assembly. HCMV may cause mild or subclinical diseases in healthy and immunocompetent adults; however, it can result in opportunistic infections in individuals whose immune system is compromised or immature [18,19], and causes multiple diseases such as pneumonitis, retinitis, gastrointestinal diseases, vascular disorder, mental retardation, mucoepidermoid carcinoma, severe cytomegalic inclusion disease of the neonate [20,21,22,23].

CMV life cycle begins as a virion attaches specific receptors on the host cell surface. For a lytic pathway, following attachment of viral envelope glycoproteins to host cell membrane receptors, the virions enter host cells by receptor-mediated endocytosis and membrane fusion. The viral capsid breaks down to release DNA genome to take over host enzyme systems to make new virions. CMV express lytic genes to demonstrate a symptomatic infection in hosts [24]. Alternatively, CMV may go into a lysogenic pathway without lysing host cells. In this pathway, CMV can exist in host cells persistently to have a latent infection, referring as the episomal latency in which viral DNA is episomal or tethered to host cell chromosomes in linear or lariat structures. The long-term latency in hosts is usually asymptomatic, though the primary infection is not necessarily asymptomatic. When CMV are stimulated by stresses or their host’s immune system is suppressed, the dormant viruses can reactivate to start producing viral progeny to cause illness, denoted as the lytic life cycle [25,26]. The mechanisms of latent-to-lytic switch can induce temporary or sustained CMV replication, causing active cytolytic inflammation. The reactivation of latent CMV can damage tissues to lead to organ diseases and may trigger indirect immunomodulatory effects to cause detrimental outcomes including increased mortality of patients and induction of graft rejection in patients undergoing organ transplantation [27].

## 2. CMV Infection

### 2.1. Congenital Infection

Because CMV can be transmitted by the close interpersonal contact such as saliva, semen, urine, breast milk, or vertical transmission (viruses pass the placenta and directly infect the fetus) [28,29] which is the leading cause of congenital infection [30,31,32,33]. Congenital CMV infection is both inherent infection and intrauterine infection [34,35]. Intrauterine transmission may occur in mothers without immune responses against CMV who are first infected by CMV in pregnancy (primary infection). Also, it may occur in the mothers with preexisting antibodies against CMV either by the reactivation of previous infections or by the infection of a different viral strain (non-primary infection) [36]. A mother can only transmit viruses to a neonate if she is undergoing primary infection or robust reactivation. Most of congenital infections are asymptomatic (85–90%) and some present with clinically apparent symptoms (10–15%), but the symptomatic disease usually emerges after primary maternal infection in pregnancy [37,38]. The morbidity risk in newborns is increased because the central nervous system (CNS) is damaged, resulting in delays in neurodevelopment, loss in hearing, and impairment in eyesight. 

### 2.2. Sequelaes

CMV infection is mostly asymptomatic among the general population; however, around 10% to 15% of infants with congenital CMV infections may be at risk of sequelaes such as mental retardation, jaundice, hepatosplenomegaly, microcephaly, hearing impairment, and thrombocytopenia [39,40,41,42]. Among these sequelaes, the most devastating one is the CNS sequelae related to neurodevelopment, because CNS injury is irreversible and persists for life, including mental retardation, seizures, hearing loss, ocular abnormalities, and cognitive impairment [43,44,45]. This means the asymptomatic newborns with congenital CMV infection still have an increased risk for long-term sequelae, especially, mental retardation and sensorineural hearing loss (SNHL) [46,47,48,49], making CMV the leading nonhereditary cause of SNHL [42,50]. 

### 2.3. Correlation between CMV Infection and Brain Diseases

CNS diseases induced by CMV infections including encephalitis and transverse myelitis are mostly found in immunocompromise patients. CMV polymerase chain reaction (PCR) assays of central spinal fluid (CSF) are available for the determination of the etiology. CMV viral loads in CSF and urine may be useful for evaluating the response for the treatment and clinical outcomes [51]. Congenital CMV infections can lead to mental retardation in newborns, but it is certainly not a main symptom of all CMV infections. 

Glioblastoma (GBM) is one of the most common and devastating brain tumors in adults [52]. Previous studies showed that some malignant GBMs express the CMV-associated proteins including the immediately early gene-1 product and the lower matrix phosphoprotein 65 (pp65) which is encoded by unique long 83 (UL83) gene, and suggested that HCMV may play an active role in glioma pathogenesis [53]. HCMV can induce GBM formation in vitro and in xenograft models. Detection of HCMV in GBM isolated from patients suggested viruses may either drive tumorigenesis or reactivate these tumors silently; however, there is no definite evidence on its ubiquity and its role in tumor formation [54]. In animal studies, murine CMV (MCMV) has been found to potentiate GBM growth in mice by increasing pericyte recruitment and angiogenesis because the set of proteins (secretome) secreted by MCMV-infected cells was changed [55]. This model provides an evidence that CMV is associated with GBM growth and supports the application of antiviral approaches for GBM therapy.

## 3. CMV Medication

Current drugs for anti-CMV infections include the inhibitors of viral DNA polymerase, such as ganciclovir, valganciclovir, cidofovir, and foscarnet. At present, ganciclovir remains to be the first line drug for the treatment of CMV infections. Ganciclovir and valganciclovir are mainly used for both preemptive therapy and prophylaxis of CMV infections in solid organ transplant (SOT) patients, but their bone marrow toxicity precludes these drugs to be used for prophylaxis after stem cell transplant [56]. Cidofovir is employed almost exclusively to treat CMV retinitis in patients with acquired immunodeficiency syndrome (AIDS) [57]. Foscarnet is principally used to treat ganciclovir-resistant CMV infections in patients with AIDS or in transplant recipients [58,59]. The combined chemotherapy (e.g., temolozomide) and radiotherapy was the standard therapy for GBM [60]. However, the prognosis is not good and only few patients survived more than 5 years. Valnoctamide, a neuroactive mood stabilizer which inhibits CMV infection in developing brain and attenuates neurobehavioral dysfunctions, was shown to have anti-CMV potential [61]. The following three novel or developing drugs (letermovir, maribavir, and brincidofovir) are also potentially applied for the treatment of brain disease associated with CMV infections.

### 3.1. Letermovir

Letermovir has been approved by the USA Food and Drug Administration (FDA) and the European Medicines Agency (EMA) for the prophylaxis of CMV infections in patients [62]. The drug was tested in CMV infected patients and may be useful for other patients who had organ transplantation or human immunodeficient virus (HIV) infections [63]. It has been clinically applied for CMV prophylaxis or treatment in hematopoietic stem cell recipients, thoracic organ recipients and lung transplantation recipients [64,65,66]. Letermovir has several advantages over conventional CMV antiviral agents such as administration orally, mild toxicity without myelotoxicity and nephrotoxicity [67,68], and targeting of the CMV terminase complex instead of CMV DNA polymerase without the possibility of cross-resistance with existing anti-CMV drugs [68]. Conventional anti-CMV drugs which target the UL54 DNA polymerase are limited in safety and efficacy because of their nephrotoxicity and viral resistance. In addition to less toxicity for kidney and bone marrow, the breakthrough of CMV treatment with letermovir is that it targets the CMV terminase complex rather than DNA polymerase, so that there is no risk to induce cross-resistance with current anti-CMV drugs. Consequently, letermovir is also promising to be used in the treatment of brain diseases associated with CMV infections.

### 3.2. Maribavir

Maribavir is a developing anti-HCMV compound which can be given orally. The drug targets the viral kinase UL97 which is crucial for the formation of viral teguments and assembly complexes for virion releasing [62]. However, it is not recommended to co-administer maribavir and ganciclovir because maribavir is an inhibitor of the UL97 enzyme which is needed for the anabolism of ganciclovir. Maribavir is commonly used when patients present with drug resistance (e.g., ganciclovir and valganciclovir) [62,69]. Additionally, it is likely to be a substitute for conventional anti-HCMV compounds because of its reduced haematotoxicity and nephrotoxicity, compared with ganciclovir and valganciclovir [62]. Maribavir has been used to treat hematopoietic cell transplant (HCT) or SOT recipients with refractory or resistant CMV infections in a clinical phase II and double-blind clinical trial [70]. However, a phase III study of maribavir in HCT and SOT recipients with refractory or resistant CMV infections is still ongoing [70,71]. 

### 3.3. Brincidofovir

Brincidofovir (CMX001) is an orally bioavailable lipid prodrug of cidofovir which is less renal toxic than the parent compound. It is an experimental antiviral CMV drug being developed for the treatment of CMV, adenovirus, smallpox, and ebolavirus infections [72]. Marty et al. significantly reduced CMV viremia in patients compared with placebo by administering brincidofovir 100 mg twice weekly (BIW) in a phase II dose-ranging trial for CMV prevention in allogeneic HCT recipients [73]. In the dose cohort, the brincidofovir 200 mg BIW only caused dose-limiting toxicity. This cohort experienced severe diarrhea and gastrointestinal acute graft-versus-host diseases. These findings promoted the performance of a safety monitoring and management proposal for the brincidofovir 100 mg BIW. Overall tolerability and safety of brincidofovir was acceptable and no dose-dependent myelotoxicity or nephrotoxicity was observed. By these results, they continued to conduct a phase III trial for the prevention of clinically significant CMV infection in allogeneic HCT recipients. However, the results showed that there was no significant reduction in the incidence of viremia among patients randomized to receive brincidofovir and this drug was still not successful in the phase III evaluation [72,73].

## 4. CMV Inhibition by Gene or RNA Targeting Approaches

External guide sequences (EGSs), transcription activator-like effectors nucleases (TALENs) and the clustered regularly interspaced short palindromic repeats (CRISPRs)/Cas9 nuclease system potentially function as effective therapeutic approaches to treat brain diseases associated with CMV infections through designing a specific DNA or RNA sequence that target genes required for viral growth. Though CRISPR/Cas9 and TALENs are both promisingly effective approaches to limit HCMV replication, for successfully used in clinical application, it is required to have accurate modification of viral genomes for avoiding off-target mutation in humans.

### 4.1. EGS-RNase

RNase P is a ribonucleoprotein complex composed of a catalytically active RNA subunit (M1 RNA) and a protein subunit (C5 protein). In addition, M1 RNA can be modified to target transfer RNA (tRNA) substrates and specific messenger RNA (mRNA) sequences. M1GS RNA, a sequence-specific ribozyme, is capable of being designed by attaching an EGS to the 3′ terminus of M1 RNA to target and cleave the specific gene sequences in cell culture and animals [74,75,76]. 

Li and Sheng et al. explored the antiviral effects of an EGS variant engineered by in vitro selection procedures using MCMV in an animal model [77]. EGSs were used to target the shared mRNA region of MCMV capsid scaffolding protein (mCSP) and assemblin which is essential for viral progeny production. In vitro, the EGS variant was 60 folds more active than the EGS originating from a natural tRNA in directing RNase P cleavage of the target mRNA. In MCMV-infected cells, the variant reduced mCSP expression by 92% and decreased viral growth by 8000 folds. In MCMV-infected mice, the EGS variant was more effective than the EGS originating from a natural tRNA in reducing mCSP expression, viral production, and mouse mortality. The results showed that engineered EGS variants with higher targeting activity in vitro are also more effective in inhibiting gene expression in vivo. Furthermore, their findings suggested its possiblility for engineered EGS variants to treat MCMV infections [77].

Li and Liu et al. used an in vitro selection procedure to engineer RNase P-based ribozyme variants with stronger targeting activity [78]. R388-AS, a novel engineered ribozyme variant, was designed to target the mRNA of assemblin of MCMV. These ribozyme variants were further used to test if they have improved activity in blocking gene expression in vivo. The results demonstrated that the engineered RNase P ribozyme variants with more active catalytic activity in vitro are also more effective in inhibiting MCMV gene expression in vivo. Additionally, their studies hinted that R388-AS can enhance ribozyme activity for therapeutic application [78].

Deng et al. revealed that engineered EGS variants induced RNase P to efficiently hydrolyze target mRNAs which encode HCMV major capsid protein [79]. In vitro, engineered EGS variants were more efficient than natural tRNA-derived EGSs in triggering human RNase P-mediated cleavage of the target mRNA by about 80 folds. In HCMV-infected cells, the EGS variant and natural EGSs led to HCMV gene expression reduction by about 98% and 73%, and the viral growth was inhibited by about 10,000 and 200 folds, respectively. The results showed that the EGS variant has improved efficiency in the inhibition of the expression of HCMV genes and viral growth, compared with the natural EGS [79].

### 4.2. CRISPR/Cas9

In the CRISPR/Cas9 system, CRISPR is used to build RNA-guided genes drives to target a specific DNA sequence. By the linkage of the specifically designed single-guiding RNA (sgRNA) and Cas proteins, the genome can be cleaved at most locations when a protospacer adjacent motif (PAM) sequence (NGG) is existing in the target site three nucleotides downstream [80]. It was also shown to successfully work as an efficient tool to engineer the genomes of a variety of organisms including HCMV [81]. Furthermore, CRISPR/Cas9 has also been considered as a potential approach applied to fight against herpesvirus infections [80]. 

Van Diemen et al. set out to combat both productive and latent HCMV infections using CRISPR/Cas9 to target genes essential for virus growth in cell culture [82]. They showed more effective inhibition of HCMV replication by targeting sgRNAs to essential viral genes UL57 and UL70 than the nonessential genes US7 and US11. The results indicated that the CRISPR/Cas9 system can be effectively targeted HCMV genomes as a potent prophylactic and therapeutic antiviral agent that may be used to block CMV replication and remove latent viruses. These new insights may allow the design of effective CRISPR/Cas9 to target HCMV during both productive and latent infections [82]. 

Gergen et al. designed two CRISPR/Cas9 systems which respectively contain three sgRNAs to target the HCMV UL122/123 gene critical for the regulation of lytic replication and reactivation from latency [83]. Both systems caused mutations in the target gene and a concomitant reduction of immediate early gene expression in primary fibroblasts. The singleplex strategy caused 50% of insertions and/or deletions (indels) in the viral genome were shown in U-251 MG cells, resulting in a decrease of immediately early protein production. The multiplex strategy cleaved 90% the immediate early gene and thereby inhibited their expression. The results revealed that viral genome replication and late protein expression were reduced by 90%. Hence, the multiplex CRISPR/Cas9 was able to target the HCMV UL122/123 gene specifically and inhibit viral replication efficiently [83].

### 4.3. TALENs

Transcription activator-like effectors (TALEs) are important virulence factors that act as transcriptional activators in the cell nucleus of plants, where they directly bind to DNA via a central domain of tandem repeats [84]. The DNA-binding domain contains a highly conserved 34-amino acid sequence, only the 12th and 13th amino acids in TALEs are variable. These two locations’ repeat variable di-residues show a strong correlation with a specific nucleotide recognition by different frequencies [85]. Currently, TALENs were shown to be an efficient tool for engineering genome precisely without toxicity and could be engineered to be an antiviral agent [86]. 

Chen et al. constructed three pairs of TALEN plasmids (MCMV1-2, 3-4, and 5-6) to target the MCMV M80 and M80.5 overlapping (M80/80.5) sequence essential for capsid assembly and further virion synthesis to test their efficacy in inhibiting MCMV growth [86]. In cell culture, the plasmids could specifically target the M80/80.5 sequence and effectively block lytic replication either using lipofectamine or a specific lipoid NKS11 as transfection reagents when TALEN plasmid transfection was prior to the MCMV infection. In animal studies, the most specific pairs of TALEN plasmids (MCMV3-4) demonstrated its capacity to inhibit the replication and gene expression of latent MCMV in immunocompetent mice using nontoxic NKS11 as transfection reagents. By the quantitative real time polymerase chain reaction (PCR) analysis, the injection of MCMV3-4 plasmids led to more significant reduction than the controls in the DNA copy number level of the immediately early gene-1 which was crucial to reactivate viruses from latency. The results suggested that TALENs potentially provided an effective approach to remove latent viruses in animals [86].

## 5. CMV Vaccine Therapy/Immunotherapy

An effective vaccine would be highly valuable in reducing human morbidity and mortality in that many diseases and complications are associated with CMV infections. The development of a safe and efficacious CMV vaccine therapy or immune therapy is paramount to the public health. However, there is still no vaccine approved for CMV [87]. The patients with GBM seldom survived five years beyond initial diagnosis and their prognosis is poor despite surgical resection, high-dose radiation, and chemotherapy with temozolomide [88,89]. Activation of the immune system against tumor cells based on vaccine therapy/immunotherapy contribute to facilitate survival and suppress tumors. CMV has emerged as an immunologic target in GBM if tumor cells have been shown to express the CMV-associated proteins immediately early protein-1 and pp65 [88,89,90,91]. The currently standard treatment does not significantly improve the clinical outcomes; thus, it is needed to develop safe and more effective therapeutic strategies such as vaccine therapy and immunotherapy.

Nair et al. explored whether T cells were stimulated by CMV pp65 RNA-transfected dendritic cells (DC) target and remove autologous GBM tumor cells [92]. CMV pp65-specific immune responses were elicited in vitro using RNA-pulsed autologous DCs recovered from patients with newly diagnosed GBM. In an antigen-specific manner, CMV pp65-specific T cells lyse autologous and primary GBM tumor cells. Moreover, T cells expanded in vitro using DCs pulsed with total tumor RNA demonstrated a 10- to 20-fold expansion of CMV pp65-specific T cells and destruction of CMV pp65-expressing target cells. The results showed that CMV-specific T cells can effectively target to kill GBM tumor cells immunologically in vitro and support the development of CMV-directed immunotherapy in patients with GBM [92]. We can conclude that immunotherapy has the potential to induce strong immune responses against brain tumor in patients with minimal impairment to surrounding normal cells and healthy tissues.

Batich et al. evaluated pp65-specific cellular responses following dose-intensified temozolomide (DI-TMZ) with pp65- DCs and determined the effects on long-term progression-free survival (PFS) and overall survival (OS) [93]. In this small randomized pilot trial, 11 patients with GBM received DI-TMZ and at least three vaccines of pp65 lysosome-associated membrane glycoprotein mRNA-pulsed DCs mixed with granulocyte macrophage colony-stimulating factors (GM-CSF) each cycle. They have previously demonstrated that targeting cytomegalovirus pp65 using DC can extend survival. Additionally, DI-TMZ and adjuvant GM-CSF induced tumor-specific immune responses in patients with GBM in another study. In this study, the results showed that pp65 cellular responses significantly increased following DI-TMZ cycle 1 and three doses of pp65-DCs. After DI-TMZ, the proportion and proliferation of regulatory T (Treg) cells were increased and continued to elevate in the following cycles. Median PFS and OS were 25.3 months and 41.1 months exceeding survival time, respectively. Though Treg cells were increased following DI-TMZ, patients receiving pp65-DCs demonstrated long-term PFS and OS, confirming their prior studies targeting CMV in GBM [93]. Overall, the results provide clinical evidence for the correlation between pp65 targeting vaccination in GBM and long-term survival and confirm that CMV pp65 represents a target resulting in GBM patients to survive longer than expected. 

Lamano et al. reported a clinical case of a woman who received immunotherapy had survived seven years post-diagnosed with GBM and recovered from CMV colitis [94]. Following tumor resection, the patient received radiation therapy and temozolomide combined therapy that was complicated by CMV colitis and abdominal abscesses. Not receiving adjuvant temozolomide, the patient still demonstrated a five-year PFS before requiring re-resection for radiation necrosis. The patient survived for two additional years following re-resection. Because the patient’s tumor was stained positive for CMV antigens of immediately early protein-1 and pp65, Thus, they had a hypothesis that the patient developed an immune response against CMV during recovery that contributed to anti-tumor surveillance and prolonged survival. In conclusion, the presented case of a GBM patient with CMV suggested the development of anti-tumor immune responses against CMV antigens that contributed to long-term survival [94]. Overall, this case report provided further insight into the role of CMV and support GBM immunotherapy. 

## 6. Conclusions

Brain diseases, a global human health problem associated with CNS, are usually complicated, serious, irreversible and difficult to treat. CMV is the leading infectious cause of mental retardation and may be associated with GBM. Currently, a variety of drugs (e.g., ganciclovir, valganciclovir, cidofovir, foscarnet, and valnoctamide) are used to treat CMV acute infections. However, their efficacy is restricted by side effects, toxicity, cross-resistance and other adverse reactions. Also, there is still no effective cure for CMV latent infections. Fortunately, some antiviral agents have been approved for clinical application (e.g., letermovir) or many therapeutic approaches are being developed (e.g., maribavir, brincidofovir, EGS-RNase, CRISPR/Cas9, TALENs, vaccine therapy/immunotherapy), though more studies and clinical trials are needed. 

For GBM treatment, the standard therapeutic approach involves maximizing safe surgical resections and followed by concurrent chemoradiation with temozolomide, but its prognosis is often poor and still universally detrimental with short survival times after initial diagnosis. Despite extensive research, only few effects and outcomes have been achieved in long-term survival of patients. Recently, the vaccine therapy and immunotherapy for GBM are areas of intense studies. Novel vaccine therapies and immunotherapies have not been extensively integrated into standard practices, though they are promising and has been applied in some clinical cases. Regardless of the possible etiologic role of CMV infections in GBM, the research interest should focus on exploiting this association for the development of vaccine therapies and immunotherapies. It is essential to investigate the interactions and correlations between CMV and hosts thoroughly to make out how therapeutic approaches influence CMV infections and improve clinical outcomes for patients. Finally, we should study the rationale and feasibility using CMV as a therapeutic target and explore the first clinical evidence for safety and efficacy using CMV-specific vaccine therapy and immunotherapy for the treatment of GBM. 

## Figures and Tables

**Table 1 ijms-21-01376-t001:** Major types of brain diseases.

Brain Disease	Brain Injury	Mental Disorder	Neurodegenerative Disease	Mental Retardation	Brain Tumor
Example	Blunt trauma	Psychoneurosis, schizophrenia, substance (e.g., drug, alcohol) addiction,	Parkinson’s disease, Alzheimer’s disease, Huntington’s disease, Spinocerebellar ataxia, Spinal muscular atrophy	Intellectual disability, general learning disability	Primary brain tumor, secondary or metastatic brain tumor, glioblastoma [11,12]
Associated factor	Hematomas, blood clots, contusions, cerebral edema, concussions, stroke	Environment, stress, genetics, physiology, neurobiochemical state	Aging, genetics	Genetics, problems at birth, malnutrition, exposure to certain diseases or toxins, cytomegalovirus (CMV) infection	Largely unclear, maybe genetics, radiation, CMV infection
Main symptom	Vomiting, nausea, speech difficulty, ear bleeding, numbness, paralysis, memory loss, concentration problem [1,2]	Depression, anxiety, agitation, delusion, stress, obsession, adaptation difficulty, emotional reaction, odd behavior, retarded intellectual development [3,4]	Memory loss, apathy, tremor, rigid, unstable posture, speech difficulty, movement retardation, facial nerve paralysis, disorientation, intellectual problem, mood change [5,6]	Delayed learning, self- care difficulty, poor planning ability, behavioral and social problems, fail to grow intellectually, fail to adapt to new situations [7,8,9,10]	Headache, seizure, numbness or tingling in arms or legs, nausea, vomiting, personality change, movement difficulty, imbalance, change in hearing, speech or vision [13,14,15]
Clinical treatment	Medication, physical therapy, language therapy, psychiatry, surgery [1,2]	Medication, psychotherapy, environmental therapy, occupational therapy, electric shock, surgery [3,4]	Medication, deep brain stimulation [5,6]	Medication, psychotherapy, language therapy, environmental therapy, occupational therapy [7,8,9,10]	Medication, surgery, chemotherapy, radiation therapy [13,14,15]

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
