# Peer review of "Potential Therapeutic Approaches Against Brain Diseases Associated with Cytomegalovirus Infections"

_ijms, 2020, doi:10.3390/ijms21041376_

Round 1

Reviewer 1 Report

This is a well written manuscript. I would suggest the following:

1) THe introduction about brain diseases is too long, and contains some general knowledge facts ("central nervous system (CNS) includes the brain, spinal cords and a large network of neurons").

2) CMV medication: I think the reader would benefit if this part was expanded (valganciclovir, 2nd line medications such as cidofovir, foscarnet, investigational drug brincidofovir).

Author Response

Reviewer 1:

This is a well written manuscript. I would suggest the following:

1. The introduction about brain diseases is too long, and contains some general knowledge facts ("central nervous system (CNS) includes the brain, spinal cords and a large network of neurons").

Ans: We have removed some statement in the introduction section which contains some general knowledge facts. (P2, line 40-42)

2. CMV medication: I think the reader would benefit if this part was expanded (valganciclovir, 2nd line medications such as cidofovir, foscarnet, investigational drug brincidofovir).

Ans: We have added more statement about valganciclovir, cidofovir, foscarnet (P.5, line 134-141) and a subsection to discussion the investigational drug brincidofovir (P.6, line 178-P7, line 92).

Reviewer 2 Report

In this review entitled, “Potential Therapeutic Approaches Against Brain Diseases Caused by Cytomegalovirus Infections”, by Wang, et al., the authors attempt to detail approaches taken to target brain disease, namely GBM, by targeting CMV. This review article merely summarizes other people’s work, with little insight from the authors on how such findings could be leveraged. One would hope that by reading such a review, that the authors would provide such additional insight, hypotheses, avenues of research, approaches, etc., that are gained from combining the current findings in the literature. But that is not done, which is rather disappointing. Further, this article suffers from a myriad of inaccuracies, which absolutely must be corrected. Details for the authors are provided below.

Major:

There are many inaccurate statements in this article that absolutely must be corrected. Line 18: “a very long-term life cycle”. CMV infection is not “very long-term”; rather it results in a life-long infection in its hosts. Table 1, Major Cause for Brain Tumor box: CMV is NOT a cause of brain tumors. There is zero evidence to support that. Lines 62-65: “…and causes multiple diseases…and other malignancies.” No. CMV is NOT the cause of ANY malignancy. Line 73: “…viral DNA is floating in the nucleoplasm…”. No. During latency, herpesviral DNA is tethered to the host cell chromosomes. It is episomal. It is not “floating” anywhere. Lines 70-71: “Alternatively, some viral genes may transcribe latency associated transcripts…”. To date, no one has identified a HCMV gene that is only transcribed during latency. All latently transcribed HCMV genes are also lytic. Thus, this is inaccurate. Lines 74-76: “The primary infection….by direct contact.” It is unclear what is meant by “limited signs” here, but primary infection will eventually always lead to latency – not “may”. Primary infection is not asymptomatic, though latency is (as worded, it is unclear if the authors meant it this way or not). Stating “…nevertheless, viruses are capable of being transmitted to other hosts by direct contact” in this same sentence is incredibly misleading. First, an individual can absolutely transmit virus via direct contact, specifically with fluids (urine, breast milk, saliva). This is not likely with a latently infected individual. However, a latently infected organ donor could transmit virus to the organ recipient. But stated as is, the authors are drastically misleading readers. Lines 85-87: Similar comment to point f above. A mother can ONLY transmit virus to a neonate if she is undergoing primary infection or robust reactivation. The authors acknowledge this later, but these lines absolutely must be altered, because as is, the information is misleading and incorrect. Line 112: “Mental retardation has been known to one of the main symptoms of CMV infections…” No. This is incorrect. Congenital CMV infections can result in mental retardation, but this is certainly not a main symptom of all CMV infections. This is inaccurate and misleading. Lines 117-118: “Therefore, mental retardation and brain tumors are likely to be resulted from CMV infections.” This is an egregious gross overstatement and must be deleted. There is no evidence brain tumors are the result of CMV infection. Linea 143-146: The authors need to review the maribavir literature. Maribavir is absolutely used in the clinics and has not failed. It was approved as a breakthrough drug by the FDA in 2018. It is commonly used when patients present with drug resistant CMV (e.g. ganciclovir, valganciclovir). This must be altered. Lines 237-238: “However, there is no licensed CMV vaccine until now.” There is no vaccine approved for CMV. Lines 295-296: “CMV is one of the etiological factors for brain diseases such as mental retardation and GBM.” This is wildly inaccurate. CMV is the leading INFECTIOUS cause of mental retardation. There is zero evidence that CMV causes GBM or any cancer for that matter. Lines 255-266: This section is confusing. What is meant by “HCMV-vectors”? Did the investigators in this study merely use a vector with the CMV promoter? It seems the work described was aimed at understanding how E6/E7, which are HPV proteins, impacted the T cell response. This would mean this study is relevant to HPV, not HCMV. While the authors summarize some, though definitely not the majority (e.g. more than IE proteins and pp65 have been implicated), of findings on HCMV’s association with GBM, they fail to discuss the opposing literature. This is necessary, as there as many studies that fail to find a connection as there are those that have found one. This must be acknowledged. The authors should really provide their own insights, interpretations, hypotheses, etc., rather than just simply regurgitating the literature. What do the authors surmise from all these details? Taking everything together, what are the authors’ thoughts? Discussion on CMV breakthrough with letermovir treatment should be noted and discussed. There are far too many reviews referenced in this review. The authors should cite primary literature, rather than reviews on the same topic others have written.

Minor

Colloquial language should be corrected. Examples: Line 67: “linking of viral envelope” (attachment of viral envelope glycoproteins) Line 68: “go into hosts” (enter host cells) Line 73: “floating in the nucleoplasm” (episomal or tethered to host cell chromosomes) Line 77: “producing lots of viral progeny” (producing viral progeny) The references for the table should be placed directly after each phrase within the table, so the reader knows which reference goes to which fact. Also, the table headings (top row, left column) should be bolded for readability. What is meant by the following phrases: Lines 18-19: “specific latent infections” Lines 69-70: “Infected cells express lytic genes to demonstrate a symptomatic infection.” Line 72: “In this pattern,…” Line 74: “accompanied by limited signs” Lines 56-60: These things are not specific to CMV. All herpesviruses have an envelope, tegument, capsid, and ds DNA genome. All herpesviruses encode their genes in a highly coordinated transcriptional cascade. Line 104: “CMV congenital infection” should be reworded to state ‘congenital CMV infection’, which is how it is referred to. Lines 113-114: GBM is the most common brain tumor IN ADULTS. Not overall. Medulloblastoma is the most common pediatric brain tumor. Lines 115-116: “pp65 which is also known as UL83”. UL83 is the gene that encodes the protein, pp65. Its not that its “also known as”.

Lines 120-124: This should be explained a bit better. Also, why is there no mention of valganciclovir (valcyte), which is incredibly common in the clinics?

Author Response

Reviewer 2:

In this review entitled, “Potential Therapeutic Approaches Against Brain Diseases Caused by Cytomegalovirus Infections”, by Wang, et al., the authors attempt to detail approaches taken to target brain disease, namely GBM, by targeting CMV. This review article merely summarizes other people’s work, with little insight from the authors on how such findings could be leveraged. One would hope that by reading such a review, that the authors would provide such additional insight, hypotheses, avenues of research, approaches, etc., that are gained from combining the current findings in the literature. But that is not done, which is rather disappointing. Further, this article suffers from a myriad of inaccuracies, which absolutely must be corrected. Details for the authors are provided below.

Major:

There are many inaccurate statements in this article that absolutely must be corrected.

Line 18: “a very long-term life cycle”. CMV infection is not “very long-term”; rather it results in a life-long infection in its hosts.

Ans: We have revised “a very long-term life cycle” into “life-long infection in its hosts”. (P1, line 19-20)

Table 1, Major Cause for Brain Tumor box: CMV is NOT a cause of brain tumors. There is zero evidence to support that.

Ans: We have revised “major cause” into “associated factor”. (P2, Table 1)

Lines 62-65: “…and causes multiple diseases…and other malignancies.” No. CMV is NOT the cause of ANY malignancy.

Ans: We have removed “and other malignancies”. (P3, line 67-68)

Line 73: “…viral DNA is floating in the nucleoplasm…”. No. During latency, herpesviral DNA is tethered to the host cell chromosomes. It is episomal. It is not “floating” anywhere.

Ans: It has been revised into “viral DNA is episomal or tethered to host cell chromosomes in linear or lariat structures.” (P4, line 77-78 )

Lines 70-71: “Alternatively, some viral genes may transcribe latency associated transcripts…”. To date, no one has identified a HCMV gene that is only transcribed during latency. All latently transcribed HCMV genes are also lytic. Thus, this is inaccurate.

Ans: We have revised the sentence into “Alternatively, CMV may go into a lysogenic pathway without lysing host cells.” (P4, line 77-78)

Lines 74-76: “The primary infection….by direct contact.” It is unclear what is meant by “limited signs” here, but primary infection will eventually always lead to latency – not “may”. Primary infection is not asymptomatic, though latency is (as worded, it is unclear if the authors meant it this way or not). Stating “…nevertheless, viruses are capable of being transmitted to other hosts by direct contact” in this same sentence is incredibly misleading. First, an individual can absolutely transmit virus via direct contact, specifically with fluids (urine, breast milk, saliva). This is not likely with a latently infected individual. However, a latently infected organ donor could transmit virus to the organ recipient. But stated as is, the authors are drastically misleading readers.

Ans: We have revised or removed the unclear, wordy and misleading sentences. (P4, line 78-81)

Lines 85-87: Similar comment to point of above. A mother can ONLY transmit virus to a neonate if she is undergoing primary infection or robust reactivation. The authors acknowledge this later, but these lines absolutely must be altered, because as is, the information is misleading and incorrect.

Ans: We have revised them into new statement. (P5, line 90-99)

Line 112: “Mental retardation has been known to one of the main symptoms of CMV infections…” No. This is incorrect. Congenital CMV infections can result in mental retardation, but this is certainly not a main symptom of all CMV infections. This is inaccurate and misleading.

Ans: We have revised this sentence into “Congenital CMV infections can lead to mental retardation in new borns, but it is not certainly a main symptom of all CMV infections.” (P5, line 121-122)

Lines 117-118: “Therefore, mental retardation and brain tumors are likely to be resulted from CMV infections.” This is an egregious gross overstatement and must be deleted. There are no evidence brain tumors are the result of CMV infection.

Ans: We have deleted this sentence. Additionally, we adjusted section 2.3 to two paragraphs to show the correlation between CMV Infection and brain diseases including mental retardation and GBM. (P5, line 123-134)

Linea 143-146: The authors need to review the maribavir literature. Maribavir is absolutely used in the clinics and has not failed. It was approved as a breakthrough drug by the FDA in 2018. It is commonly used when patients present with drug resistant CMV (e.g. ganciclovir, valganciclovir). This must be altered.

Ans: We have added two sentences including references as follow: “Maribavir is commonly used when patients present with drug resistance (e.g., ganciclovir and valganciclovir) [62, 69].” (P6, line 171-172), “However, a phase III study of maribavir in HCT and SOT recipients with resistant or refractory CMV infections is still ongoing [70, 71].” (P6, line 178-179)

Lines 237-238: “However, there is no licensed CMV vaccine until now.” There is no vaccine approved for CMV.

Ans: We have revised the sentence into “There is still no vaccine approved for CMV” (P9, line 287-288)

Lines 295-296: “CMV is one of the etiological factors for brain diseases such as mental retardation and GBM.” This is wildly inaccurate. CMV is the leading INFECTIOUS cause of mental retardation. There is zero evidence that CMV causes GBM or any cancer for that matter.

Ans: We have revised the sentence into “CMV is the leading infectious cause of mental retardation and may be associated with GBM.” (P10, line 351-P11, line 352)

Lines 255-266: This section is confusing. What is meant by “HCMV-vectors”? Did the investigators in this study merely use a vector with the CMV promoter? It seems the work described was aimed at understanding how E6/E7, which are HPV proteins, impacted the T cell response. This would mean this study is relevant to HPV, not HCMV. While the authors summarize some, though definitely not the majority (e.g. more than IE proteins and pp65 have been implicated), of findings on HCMV’s association with GBM, they fail to discuss the opposing literature. This is necessary, as there as many studies that fail to find a connection as there are those that have found one. This must be acknowledged.

Ans: In this article we cited, the authors designed novel HCMV-based therapeutic viral vaccines to exploit the patient’s own immune system for elimination of tumor cells. Finally, they tested whether genetically altered T cells specific for HCMV encoded epitope or neo-epitope (E6/E7 which are HPV proteins) are stimulated by GBM cells infected with the HCMV-based vaccines. It seems the work described was aimed at understanding how E6/E7 impacted the T cell response. This mean this study is relevant to HPV, not closely related to HCMV. Therefore, we have removed this section. (P9, line 308-P10, line 319)

The authors should really provide their own insights, interpretations, hypotheses, etc., rather than just simply regurgitating the literature. What do the authors surmise from all these details? Taking everything together, what are the authors’ thoughts?

Ans: We have tried our best to added some statement to provide our own insights, interpretations, hypotheses, etc. in every section. (P6, 7, 9, 10)

Discussion on CMV breakthrough with letermovir treatment should be noted and discussed.

Ans: We have added some statement to note and discuss the CMV breakthrough with letermovir treatment. (P6, line 160-164)

There are far too many reviews referenced in this review. The authors should cite primary literature, rather than reviews on the same topic others have written.

Ans: We have tried our best to added more primary literatures and cite them in this manuscript (Reference number has increased from 84 to 94).

Minor

Colloquial language should be corrected. Examples:

Line 67: “linking of viral envelope” (attachment of viral envelope glycoproteins)

Ans: We have revised it into “attachment of viral envelope glycoproteins”. (P4 line 70-71)

Line 68: “go into hosts” (enter host cells)

Ans: We have revised it into “enter host cells”. (P4, line 71)

Line 73: “floating in the nucleoplasm” (episomal or tethered to host cell chromosomes)

Ans: We have revised it into “episomal or tethered to host cell chromosomes”. (P.4, line 78)

Line 77: “producing lots of viral progeny” (producing viral progeny)

Ans: We have revised it into “producing viral progeny”. (P4, line 82)

The references for the table should be placed directly after each phrase within the table, so the reader knows which reference goes to which fact. Also, the table headings (top row, left column) should be bolded for readability.

Ans: We have placed the references directly after each phrase within the table 1. (P2-3)

What is meant by the following phrases?

Lines 18-19: “specific latent infections”

Ans: It has been revised into “latent infections”. (P1, line 20)

Lines 69-70: “Infected cells express lytic genes to demonstrate a symptomatic infection.”

Ans: It has been revised into “CMV expresses lytic genes to demonstrate a symptomatic infection in hosts.” (P4, line 73-74)

Line 72: “In this pattern,”

Ans: It has been revised into “ In this pathway,”. (P4, line 76)

Line 74: “accompanied by limited signs”.

Ans: It has been revised into “The long-term latency in hosts is usually asymptomatic, though the primary infection is not necessarily asymptomatic.” (P4, line 78-80)

Lines 56-60: These things are not specific to CMV. All herpesviruses have an envelope, tegument, capsid, and ds DNA genome. All herpesviruses encode their genes in a highly coordinated transcriptional cascade.

Ans: It has been revised into “Like all herpesviruses, HCMV encodes it genes in a highly coordinated transcriptional cascade and gene expresses in an order as follows:” (P3, line 59-61)

Line 104: “CMV congenital infection” should be reworded to state ‘congenital CMV infection’, which is how it is referred to.

Ans: It has been revised into “congenital CMV infection”. (P4, line 106)

Lines 113-114: GBM is the most common brain tumor IN ADULTS. Not overall. Medulloblastoma is the most common pediatric brain tumor.

Ans: It has been revised into “ GBM is one of the most common and devastating brain tumors in adults. “ (P5, line 123-124)

Lines 115-116: “pp65 which is also known as UL83”. UL83 is the gene that encodes the protein, pp65. It’s not that its “also known as”.

Ans: We have revised the sentence into “phosphoprotein 65 (pp65) which is encoded by unique long 83 (UL83) gene” (P5, line 124-127)

Lines 120-124: This should be explained a bit better. Also, why is there no mention of valganciclovir (valcyte), which is incredibly common in the clinics?

Ans: We have added some statement to mention valganciclovir. (P3, line 136-143)

Reviewer 3 Report

This is a good overview of the role of CMV in human brain disorders. The review discusses multiple factors including antiviral strategies, and multiple diseases including congenital CMV and glioblastoma.  As an overview, I believe it does a good job. 

Minor spelling issues:

pg.3 line 57 should be "HCMV genes are expressed..."

pg. 8 line 246 should be "T cells were stimulated"

Author Response

Reviewer 3:

This is a good overview of the role of CMV in human brain disorders. The review discusses multiple factors including antiviral strategies, and multiple diseases including congenital CMV and glioblastoma.  As an overview, I believe it does a good job. 

Minor spelling issues:

pg.3 line 57 should be "HCMV genes are expressed..."

Ans: It has been revised into “HCMV genes are expressed…..” (P3, line 59-61)

pg.8 line 246 should be "T cells were stimulated"

Ans: It has been revised into “T cells were stimulated". (P9, line 297)